# A Review of Omics Studies on Arboviruses: Alphavirus, Orthobunyavirus and Phlebovirus

**DOI:** 10.3390/v14102194

**Published:** 2022-10-05

**Authors:** Rafaela dos S. Peinado, Raphael J. Eberle, Raghuvir K. Arni, Mônika A. Coronado

**Affiliations:** 1Multiuser Center for Biomolecular Innovation, Department of Physics, Sao Paulo State University, Sao Jose do Rio Preto 15054-000, SP, Brazil; 2Institute of Biological Information Processing, IBI-7: Structural Biochemistry, Forschungszentrum Jülich, 52425 Jülich, Germany; 3Institut für Physikalische Biologie, Heinrich-Heine-Universität, 40225 Düsseldorf, Germany

**Keywords:** arboviruses, omics, genomics, transcriptomics, proteomics, metabolomics

## Abstract

Since the intricate and complex steps in pathogenesis and host-viral interactions of arthropod-borne viruses or arboviruses are not completely understood, the multi-omics approaches, which encompass proteomics, transcriptomics, genomics and metabolomics network analysis, are of great importance. We have reviewed the omics studies on mosquito-borne viruses of the *Togaviridae*, *Peribuyaviridae* and *Phenuiviridae* families, specifically for Chikungunya, Mayaro, Oropouche and Rift Valley Fever viruses. Omics studies can potentially provide a new perspective on the pathophysiology of arboviruses, contributing to a better comprehension of these diseases and their effects and, hence, provide novel insights for the development of new antiviral drugs or therapies.

## 1. Introduction

Arboviruses form a diverse group consisting of over 500 viruses from different families that are transmitted by a variety of arthropod vectors, including mosquitoes, ticks, flies and midges, where they multiply, resulting in a high-titred load, especially in salivary glands, and are subsequently transmitted to humans and other vertebrate hosts through bites. Their transmission can be maintained through a sylvatic cycle, where viruses are stably transmitted from vectors, and wild animals with sporadic spillover to humans or domestic animals, or a human or domestic animal host infected can act as an amplifier to other humans and animal hosts and trigger epidemics, which is the case of the urban cycle [1]. Their emergence and re-emergence are of great public health importance. The clinical effects are associated with neurological, viscerotropic and hemorrhagic diseases. Their ability to adapt to new environments, new arthropods, and human hosts causes major health and socioeconomic issues. The great majority are classified as neglected diseases due to the lack of antiviral treatment, with only symptomatic treatments and prophylactic measures to contain the available vectors [2,3].

Considering the impact that arboviruses have throughout the world, a multi-omics approach to comprehend viruses, virus-host interactions, and the disease process is of significant relevance. Genomics, transcriptomics, proteomics and metabolomics have been widely used as technological advances made high-throughput and cost-efficient analysis available. From the identification of genetic variants of the diseases, characterization, RNA (transcript) levels, protein abundance, structure and interactions to the identification of small molecules derived from cellular metabolism, the disease process can be deciphered as well as disease biomarkers, which are of utmost importance for the development of future antiviral therapies and drugs [4]. Technological advancements in techniques such as DNA sequencing [5], transcriptomics via RNA-seq [6], SWATH (Sequential Windowed Acquisition of all Theoretical fragments) and Mass Spectrometry (MS)-based proteomics [7,8], Nuclear Magnetic Resonance (NMR), and MS-based metabolomics [9,10] permit us to integrate data from these different areas, although still challenging, as the specific analytical tools for each omics must be well-suited for logical comparisons of the obtained results [11].

Here, we present a review of general viral characteristics along several omics studies applied to well-known mosquito-borne viruses within the *Togaviridae*, *Peribuyaviridae* and *Phenuiviridae* families, more specifically, the following arboviruses: Rift Valley Fever, Oropouche, Chikungunya and Mayaro virus [12,13].

## 2. An Overview of the Genera

### 2.1. Togaviridae—Alphavirus

Arboviruses from the *Alphavirus* genus are classified into three major clades, including several complexes. The Venezuelan equine encephalitis complex comprises the Venezuelan Equine Encephalitis virus (VEEV), and the Western Equine Encephalitis complex comprises the Western Equine Encephalitis virus (WEEV) and the Sindbis virus (SINV). Another complex is the Semliki Forest complex, in which we find the Semliki forest virus (SFV), Chikungunya virus (CHIKV), Mayaro virus (MAYV) and Ross River virus (RRV). Other known complexes are the Barmah Forest, the Eastern equine encephalitis complex, and the Middelburg and Nmdumu complexes [14]. *Alphaviruses* belong to the *Togaviridae* family, characterized as enveloped, spherical viruses with a single positive RNA strand. The *Alphavirus* viral particle is around 70 nm in diameter T = 4 icosahedral symmetry capsid, which presents 240 copies of the capsid protein in the nucleocapsid core (NC), coated by a host-derived lipid membrane with glycoprotein spike-like projections that consist of envelope E1 and E2 heterodimers associated as trimers [15]. The ~11 kb single-stranded RNA codes for structural and nonstructural proteins, as illustrated in Figure 1 [16], and contains a 5’ 7-methyl-GpppA cap and a 3’ poly(A) tail [17]. Genome organization can be summarized as 5′m^7^G-nsP1-nsP2-nsP3-nsP4-(junction)-C-E3-E2-6K/TF-E1-An3′ [18]. Some conserved and repeated sequences are present in the 5’ and 3’ ends, which are important for RNA synthesis and replication regulation. Two open reading frames (ORFs) are present in the genome. The first ORF is responsible for translating directly from genomic RNA and encodes the four nonstructural proteins (nsP1, nsP2, nsP3 and nsP4) required for RNA synthesis [19]. The second ORF is expressed by producing a subgenomic mRNA generated by an internal promoter in the minus-strand RNA replication intermediate. It encodes the structural proteins C—capsid, E—envelope E3, E2, E1 and 6K/TF (transframe is a C-terminal extension of the 6K protein after a frameshift event), which are responsible for the assembly of new virion particles and their entry into the cells. The short 5’ and 3’ untranslated regions form stem-loop structures with repeats are presumed to be responsible for viral adaptation to a range of vectors and hosts. They are presumably responsible for viral gene expression, replication, and virus-host interactions [20]. The capsid protein (C) is the icosahedral nucleocapsid (NC) component and is present with 240 copies that compose the important structure for viral RNA packaging. The NC is formed by a host-derived lipid-bilayer where E1 and E2 glycoproteins are embedded along with smaller amounts of other membrane-associated proteins (6K/TF) that also form part of the viral particle [21]. As E1 and E2 interact, a rigid structure is formed across the viral membrane, creating the icosahedral format that surrounds the virus. E3 glycoprotein works as a signal sequence for insertion of the remaining polyprotein into the endoplasmic reticulum for later processing, and 6K seems to act as a signal sequence for processing the E1 protein [22,23].

Viral entry in cells is mediated mainly by clathrin proteins through membrane invagination and scission to form a clathrin-coated vesicle that contains the viral particle, which is delivered into the cell and taken by endosomes with a low pH environment that facilitates E1/E2 glycoproteins fusion to the endosomal membrane due to putative binding sites (Figure 2). Multiple entry receptor pathways have been suggested, including membrane proteins prohibitins (PHB) [25], phosphatidylserine-mediated virus entry-enhancing receptors (PVEERs) [26], glycosaminoglycans (GAGs) [27], and others [28,29]. After fusion, the nucleocapsid is released and dismantled in the cytoplasm, and the viral genome is liberated (Figure 2). The translation of the RNA genome is a membrane-associated process that, in most *Alphavirus* and RNA viruses, induces host membranes into forming cytoplasmic structures known as type-1 cytopathic vacuoles (CPVs). The genomic RNA is translated into the P1234 polyprotein, which is then cleaved by nsP2 protease function and forms the minus strand replicase complex P123 and nsP4 [30]. This complex initiates the minus strand RNA synthesis that will be the template for the production of genomic and subgenomic RNAs. The P123 polyprotein is also cleaved into nsP1 and P23, forming the nsP1/P23/nsP4 replicase complex that synthesizes positive-strand RNAs from the minus strand RNAs previously produced for genomic and subgenomic production [31].

After this process, the P23 portion, which is present only for a few seconds, is processed, yielding the fully cleaved replicase complex that promotes RNA synthesis with the individual nonstructural proteins nsP1, nsP2, nsP3 and nsP4 [19,33]. After forming the RNA replicase complex by the nonstructural proteins, the positive-stranded RNA is translated into a polyprotein that is cleaved into three structural proteins: C—the capsid protein, PE2, and E1. PE2 is then processed into E2 and E3, responsible for envelope assembly together with the E1 glycoprotein. The viral envelope comprises a lipid bilayer derived from the host cell membrane and the main E1 and E2 glycoproteins anchored as spikes in an orderly arrangement [34,35]. The structural proteins, essential for virion assembly, are translated from subgenomic RNA, as CHIKV shuts down cell mRNA translation to produce its mRNAs. Viral particle assemblage occurs on the membrane of infected cells, and nsP1 works as a defence mechanism against host antiviral effects at this stage [29]. As alphaviruses disseminate mostly through the lymphatics and microvasculature, the blood is loaded with virion particles and infected monocytes that target other organs. The liver and spleen are sites for further viral replication and dissemination (Figure 3). Subsequently, the viral particles can reach bones, muscles, and articular tissues, which characterizes the acute phase of the disease, where inflammatory infiltrate rich in monocytes, macrophages, natural killer cells, CD4+ and CD8+ lymphocytes are prevalent and affect joints and muscles, inducing the characteristic arthralgia and arthritis symptoms [36,37,38].

Regarding the nonstructural proteins, nsP1 presents both guanine-7-methyltransferase and guanylyl transferase activities required for the capping and methylation of new viral genomic and subgenomic RNA. It is also believed to anchor replication complexes to the cytoplasmic membrane, which is required to initiate and maintain the minus-strand replicative intermediates [22], along with membrane and cytoskeletal rearrangements [39,40]. The nsP2 protein possesses several enzymatic functions in its N-terminal regions, such as helicase, nucleoside triphosphatase (NTPase), and RNA-dependent 5′-triphosphatase activity. In contrast, the C-terminal region yields a viral cysteine protease required for nonstructural polyprotein processing [41]. Also, during infection, a portion of nsP2 remains in the nucleus. It is responsible for host-cell transcription shut-off and cytopathogenicity [42,43]. Along with E1 and E2, it has been shown to strongly antagonise IFN-β and other signalling molecules involved in the MDA5/RIG-I receptor signalling pathway [44].

Concerning nsP3, it is known to present three domains: the macro domain, the alphavirus unique domain (AUD), and the hypervariable region (Figure 1) [19]. The N-terminal region of nsP3 is highly conserved among alphaviruses and has a macrodomain that presents homologues across other domains of life [45]. The macrodomain has nucleic acid binding and phosphatase activities and has been found to bind to DNA, RNA and poly ADP-ribose [46,47]. The AUD comprises the central portion of nsP3 and has strong sequence homology among alphaviruses, and the C-terminal domain, which is the hypervariable region, presents poor conservation among the genus [48]. nsP3 associates with other nsPs as part of the replication complex and is required for replication due to its N-terminal region, which also presents an ADP-ribose 1-phosphate phosphatase activity region [24,49]. It is known that some part is also liberated and forms cytoplasmic aggregates that can interact with host factors such as G3BP (Ras-GTPase-activating protein (SH3 domain)-binding proteins) proteins and others, independent of but possibly enhancing replication [24,50]. In some alphaviruses, nsP3 can also act as an important neurovirulence factor [51], although E2 and other structural proteins are usually virulence determinants [24,52].

Finally, nsP4, the highly conserved polymerase among alphaviruses [53], is suggested to act as a scaffold for interaction with other host proteins or nsPs. When not integrated into the replication complex, it is targeted for degradation by a ubiquitin-dependent N-end rule pathway, depending on the N-terminal residue [54]. Its N-terminal domain presents adenylyl transferase activity (ATase), possibly indicating a polyadenylation function [55]. In contrast, the C-terminal region has an RNA-dependent RNA-polymerase (RdRp) function which is solely responsible for the RNA synthetic function of the replication complex associated with the P123 replication complex after the P1234 polyprotein cleavage [56,57].

### 2.2. Bunyavirales—Orthobunyavirus and Phlebovirus

The *Orthobunyavirus* (Oropouche Virus) and *Phlebovirus* (Rift Valley Fever Virus) genera belonged to the *Bunyaviridae* family until 2016, when the International Committee on Taxonomy of Viruses (ICTV) created the order *Bunyavirales* that replaced *Bunyaviridae* and placed the *Orthobunyavirus* and *Phlebovirus* into the *Peribuyaviridae Phenuiviridae* families, respectively [58]. These viruses are negative or ambisense RNA viruses that can infect a variety of animals, plants, and humans with the capacity to cause severe disease in hosts, including haemorrhagic fever, with over 350 viruses mainly transmitted by arthropod vectors [59,60]. They are lipid-enveloped viruses of 80 to 120 nm in diameter that include three ssRNA segments: large (L), medium (M) and small (S). These segments are surrounded by a helicoidal nucleocapsid and encode the RNA-dependent RNA polymerase (RdRp), the surface glycoproteins Gn and Gc arranged in spikes, and the nucleocapsid protein N, along with two other nonstructural proteins NSm and NSs that are encoded by the M and S segment, respectively, as shown in Figure 4 [60,61,62], which are mainly associated with virulence. The coding sequences of the three segments are flanked by two terminal non-translated regions (NTRs) 5’ and 3’ that present 11 nucleotides, highly conserved among the segments but with a variable number of nucleotides in length. Such regions are complementary, allowing genomic RNA circularization to work as replication and transcription promoters [63,64].

The small (S) RNA segment encodes the N protein, which presents regions involved in oligomerization, polymerase binding and RNP (ribonucleoprotein complex) assembly into virus particles and is also thought to promote virus template recognition by the RNA-dependent RNA polymerase [66]. The S segment also encodes the nonstructural protein NSs responsible for regulating host antiviral response by inhibiting host transcription and terminating interferon production, therefore working as a major virulence factor. Both proteins are translated from the same mRNA produced by the S segment [67]. The medium (M) segment yields a polyprotein precursor inserted into the endoplasmic reticulum membrane during translation. Later, it is cleaved by host proteases into two polypeptides called Gn and Gc [66,67]. Together they form the Gn-Gc heterodimer, which is retained in the Golgi apparatus due to glycosylation and promotes virus assembly associated with the RNP complex, viral particle formation, budding, attachment and entry into new host cells. Most orthobunyaviruses also produce the NSm protein from the same polyprotein that serves as a complex for translation initiation and polyprotein processing [68]; it is thought to participate in viral assembly as well as inter-cellular transmission. At the same time, for *Phlebovirus*, NSm is non-essential for replication, but there is speculation that it may work on apoptosis regulation mechanisms [69].

The viral RNA-dependent RNA polymerase (RdRP) is encoded by the large segment (L), the largest segment with ~250 kDa, for RNA replication and mRNA transcription. A comparison of several RdRPs from negative-strand RNA viruses shows a well-known polymerase module, including its motifs pre-A, A, B, C, D and E [68,70]. The RNA segments are templates for RNA replication and mRNA transcription. They are always associated with viral NPs (nucleoprotein complex) with at least one copy of the L protein assembled into the complex due to the negative polarity of the viral RNA (vRNA), and the one needed first to transcribe their mRNAs and produce the viral proteins essential for replication and formation of new virion particles [65]. The synthesis of a complementary RNA (cRNA) with positive polarity starts with 5’ nucleoside triphosphates for genomic replication, and it is the viral RNA segment’s full-length complementary (antigenomic) copy. At the same time, transcription requires a host-cell derived cap primer to signalize initiation and ends at specific termination signals before the 3’ end region without adding a poly-(A) tail [66,71,72].

Viral entry in the cell generally occurs through a clathrin-mediated endocytic pathway, with different pathways and interactions that have been proposed for different bunyaviruses, including members of the integrin family, and binding to filopodia or dendrites with the formation of non-coated vesicles [60] and glycosaminoglycans. After membrane fusion and endocytosis, the viral ribonucleoprotein complex (RNP) is released into the cytoplasm. It accumulates in the Golgi stacks, where so-called viral tubes are formed, composed of the host cell and viral components (NSm) [73], for the viral polymerase packed with the RNP to initiate primary transcription for the synthesis of mRNA. During the primary transcription, N mRNA and NSs mRNA are produced (~40 min after infection) from the S segment. Within two hours post-infection, replication of the viral RNA commences, increasing mRNAs and proteins. The RNP is probably packed into the virions by interaction with the cytoplasmic domains of Gn and Gc at the Golgi apparatus. The three RNA segments are co-packed in a manner where M and S segments may assist L segment packaging [74]. Finally, the new virions bud from the Golgi complex is released from the cells undergoing further morphological changes for maturation and total infectivity capacity (Figure 5) [60].

## 3. Omic’s Study across Members of the Togaviridae Family

### 3.1. Chikungunya Virus

Chikungunya virus (CHIKV) is the pathogen responsible for Chikungunya fever, mainly transmitted by *Aedes aegypti* and *Aedes albopictus* mosquitoes. It was first recognised in an epidemic on the Makonde Plateau in the Southern Province of Tanganyika (Tanzania) from 1952 to 1953 [75], and since then has triggered several outbreaks and epidemics around the world and has been identified in over 40 countries [76,77]. The word “Chikungunya” belongs to the Makonde dialect and means “that which bends up”, which refers to the contorted posture of infected patients due to severe joint pain. The clinical manifestations of CHIKV may vary but are usually very debilitating and are characterized by sudden onset fever that can appear two days post-infection, skin rash, arthralgia and arthritis, joint and muscle pain, joint swelling, fatigue, nausea, vomiting, headaches and even conjunctivitis. Infected individuals may present protracted arthralgia even weeks, months, or years post-infection. Some cases observed from the 2006 outbreak on La Réunion Island included encephalitis, Guillain-Barré syndrome, myocarditis, and hepatitis [78].

It is known that the CHIKV lineages spread across the world derive both from African and Asian lineages (the Asian Urban lineage—AUL; the Indian Ocean lineage—IOL; the East, Central and South African lineage—ECSA; and the West African lineage—WA) in a series of genetic and evolutionary events that enabled adaptation and transmission in new environments and vector species. A recent investigation of CHIKV phylogeny with analysis of the RNA architecture of the UTRs (untranslated regions) from all CHIKV lineages, which are well-conversed evolutionarily with known elements related to viral pathogenesis, revealed some lineage-specific structured and unstructured repeat elements that are possibly involved in pathogenicity, vector adaptation and specificity, viral replication and host factor binding [79,80]. Based on the results and geographic distribution of CHIKV lineages, the authors suggested three major epidemic clades, the ECSA-MASA (ECSA + Middle Africa/South American lineages), the AUL-Am (Asian Urban + American lineages), and the ECSA-IOL (Eastern Africa + Indian Ocean Lineage). The main observation was that the ECSA lineage is no longer a single lineage due to the co-circulation of multiple lineages revealing the presence of ECSA subgroups. Composition and length of the 3′UTRs were also compared, revealing substantial differences among lineages and the characterization of five highly conserved regions into four structured and one non-structured element, including stem-loop elements. However, their functional roles remain to be determined [80].

CHIKV virus infection generates temporary dsRNA intermediates during the replication process that can engage with pathogen receptors in cells such as Toll-like receptor 3 (TLR3), TLR7, TLR8 and the retinoic-acid-inducible gene I (RIG-I) [81,82], which induces a cascade leading to the activation of type I interferons (IFNs), cytokines and chemokines production. IFNs activation is regulated by MAVS (mitochondrial antiviral-signalling protein), and IL-1β production may be induced. In mouse models, the defence mechanism was also dependent on the TLR adaptor myeloid differentiation primary response protein 88 (MYD88), which can also act as an adaptor for the interleukin-1β receptor (IL-1R) that may be activated by the secretion of IL-1β from infected cells and activate type I IFN in cells that are not infected [77,83]. Moreover, it has been reported that CHIKV nsP2, E2 and E1 proteins are major suppressors of IFN-β activation through the MDA5-RIG-I pathway, and nsP2 has been specifically shown to induce host cell shut-off by inhibiting cellular transcription and therefore promoting a decrease in signalling molecules involved in the IFN-β pathway to effectively evade host immunity (Figure 6) [43,84].

Mammalian and human cells infected with CHIKV rapidly undergo apoptosis. It presents highly cytopathic effects due to the disturbance of cell physiological processes, inhibiting cell transcription and translation and redirecting cellular mechanisms and resources towards viral replication and virion production. However, innate immune response with type I and II IFNs can usually reduce viral content within a few days post-infection, reducing symptoms and viremia [85]. Numerous proinflammatory mediators are produced during infection, including IFN-α, IL-4, IL-10 and IFN-γ, with a high circulation of CD8+ and CD4+ lymphocytes, monocytes and leukocytes. During the early phases of the infection, infected monocyte/macrophages migrate to the synovial tissues of infected patients, contributing to the inflammation process and joint pain symptoms, which may persist for a long time after infection because these areas may become viral reservoirs, generating chronic arthralgia. In addition, these infected cells may also be responsible for viral dissemination in other tissues, such as the nervous system, and contribute to the development of other clinical features [86].

Using RNA-seq and Ribo-seq, Jungfleisch and colleagues showed that CHIKV infection induces codon-specific reprogramming of the host translation machinery in order to favour the translation of viral RNA genomes over host mRNAs. This reprogramming was mostly apparent at the endoplasmic reticulum, where CHIKV RNAs showed high ribosome occupancy. Mechanistically, it involves the CHIKV-induced overexpression of KIAA1456, an enzyme that modifies the wobble U34 position in the anticodon of tRNAs, which is required for the proper decoding of codons that are highly enriched in CHIKV RNAs [87].

Recently, defective viral genomes (DVGs) generated during infection in mammalian and mosquito cells revealed interesting antiviral effects that interfere with viral dissemination, especially in *Ae. aegypti* cells. It was shown that, although DVGs vary in type and abundance depending on the viral strain and host environments, most DVGs possessed promising inhibiting abilities resulting from the hijacking of the replication machinery and host resources in a competitive manner against non-defective genomes. Some DVGs may even function against other CHIKV strains as well as other alphaviruses. Such cross-reactivity may be used in favour of antiviral research for the development of future antiviral studies and therapies against arthritogenic alphaviruses. However, much research is still needed on identifying these DVGs and testing their antiviral functions, as well as developing efficient delivery systems [88]. Another inducer of antiviral response discovered in a recent transcriptomic RNA-seq analysis was IL27 (interleukin 27). The results of this study demonstrated that CHIKV-infected human monocytes-derived macrophages (MDMs) with recombinant expressed IL27 lead to the activation of JAK-STAT signalling, inducing a pro-inflammatory and antiviral response in order to control viral replication. Furthermore, treatment of cells with IL27 inhibits CHIKV in a dose-dependent manner [89].

Proteomic analyses of liver and brain tissues from mock and CHIKV infected newborn mice by two-dimensional electrophoresis (2-DGE) revealed differentially expressed proteins involved in iron metabolism, with upregulated transferrin levels (mostly related to defence against infection), dysregulation of the urea cycle and fatty acid oxidation. Furthermore, many proteins identified were related to cellular stress, such as catalase, peroxiredoxin-6, molecular chaperone proteins (Hspd1 and DNA K chaperonin), and heat shock proteins (HSPs), all related to stress response, including oxidative stress, inflammation and apoptosis. Other related pathways affected by CHIKV were energy metabolism (decrease in glycolytic enzymes), urea cycle enzymes (carbamoyl phosphate synthase, arginase, ornithine carbamoyltransferase, glutamate dehydrogenase—leading to accumulation of ammonia). CHIKV infection results in the rise of nitrogenous metabolites, cytoskeletal proteins (coronin 1A), fatty acid metabolism (increased levels of apolipoprotein A-IV and decreased Acox1), apoptosis (with upregulated levels of prohibitin) and overall inflammation in both tissues, with a clear demonstration of CHIKV neurotropism and neuroinvasiveness potential [90]. Comparatively, proteomic profiles of WRL-68 cells (a human hepatic cell line) by 2-DGE and MALDI-TOF/TOF (matrix-assisted laser desorption/ionization time-of-flight mass spectrometry) during early infection by CHIKV revealed alteration in the expression of proteins related to mRNA processing and translation, energy and cellular metabolism, ubiquitin-proteasome pathway (UPP), and cell cycle regulation, with cyclin-dependent kinase 1 (CDK1) regulation for efficient viral replication, survival and transmission [91].

Treffers and collaborators, using SILAC LC/MS/MS (Liquid Chromatography Tandem Mass Spectrometry) in a temporal approach—8, 10 and 12 h post-infection, identified over 4700 proteins, and the majority showed decreased abundance over time [92]. In agreement to host shut-off caused by the virus, RNA polymerase II complex subunits presented progressive degradation. Four other proteins, namely Rho family GTPase 3 (Rnd3), DEAD box helicase 56 (DDX56), polo-like kinase 1 (Plk1) and ubiquitin-conjugating enzyme E2C (UbcH10), which are likely overexpressed in order to reduce cell susceptibility to CHIKV infection, also presented downregulation during infection as a mechanism for efficient viral replication [92]. Moreover, in a 2-DGE proteomic analysis combined with MALDI-TOF MS of human muscle cells, the expression of proteins involved in the synthesis and metabolism of biomolecules, cell signalling, reorganization, cellular stress, and gene silencing via siRNA was observed. Interestingly, the authors also observed an interaction of CHIKV nsP3 with vimentin filaments, suggesting that this association plays an essential role in the anchorage and trafficking of replication complexes (RCs) to ensure efficient viral replication [93].

A recent TimsTOF Pro Based LC/MS/MS proteomic study of *Ae. aegypti* mosquitoes with the oral acquisition of CHIKV demonstrated exquisite but minor changes in the mosquito proteome, possibly due to the adaptation of the vector to the infection. Overall, enrichment of pathways such as oxidative phosphorylation, endocytosis and ribosome biogenesis was observed. At the same time, the attenuation of cellular RNA machinery related to RNA transport, RNA polymerase and Aminoacyl-tRNA biosynthesis was reported, possibly for viral RNA protection from degradation in the mosquito and for maintaining infection, or even for host defence as a means to decrease viral replication and consequent titer yields in cells. Likewise, cytoskeleton proteins such as syntenin, actin, twinfilin and the vesicle trafficking MON1-CCZ1 complex-like protein were modulated by infection, with decreased levels as infection progressed (from 24 h post-infection onward), possibly as a host response to avoid de novo cell and tissue infection. Finally, immune-related proteins (RNA interference—RNAi; immune deficiency factor—IMD; Janus kinase-signal transducer and activator of transcription—JAK-STAT; and defensin C) and serine-type endopeptidases and metalloproteinases related to cell entry, signalling, viral synthesis, maturation and release were also affected during infection, although further studies are essential to properly assess their functions related to the infection itself [94].

An isobaric tag-based high-resolution MS (mass spectrometry) proteomic study of infected *Ae. aegypti* salivary glands by CHIKV, comparing the effects of other two arboviruses (Dengue and Zika virus), showed virus-specific regulation patterns related to immunity, blood-feeding and cellular machinery processes, with a total of 58, 27 and 29 regulated proteins (by Dengue, Zika and Chikungunya viruses, respectively) and four upregulated proteins common to the three infections all related to antiviral functions (ADA—adenosine deaminase, SGS1—salivary gland surface protein 1, GILT-like protein—gamma-interferon responsive lysosomal thiol-like protein, and SGBAP—salivary-gland broad-spectrum antiviral protein). Results showed that the ADA, a saliva protein that was upregulated in all three infections, exerted an anti-CHIKV effect, possibly through regulating immune activation and repression. The SGBAP had an antiviral potential against all three viruses by inhibiting viral propagation. However, its mechanisms remain to be discovered through homily investigation [95].

Moreover, a very recent label-free quantitative proteomic analysis of *Ae. aegypti* Aag-2 cells revealed a total of 196 regulated proteins related to energy metabolism, protein synthesis, signalling pathways and apoptosis, with antiviral and proviral effects. A known mechanism to control viral propagation was observed with the shutoff of general protein synthesis in a 48 h post-infection period and the induction of autophagy by ROS (Reactive oxygen species) production in mitochondrial nets to delay apoptosis and prolong survival of the infected cells before apoptosis [96].

In order to comprehend the mechanisms involved in CHIKV-induced arthralgia, a proteomic analysis of primary human fibroblast-like synoviocytes (HFLS), which play a pivotal role in joint damage in arthritic disorders, was performed through gel-enhanced chromatography-mass spectrometry (GeLC-MS/MS). The study was conducted 12 and 24 h post-infection, and 259 and 241 proteins were identified, respectively. Investigation of protein functions revealed mechanisms such as cytoskeletal organization (MAEA—macrophage erythroblast attacher; NEFL—neurofilament light polypeptide and FLG—filaggrin) and cellular trafficking (Rab-8B—Ras-related protein; DNAH5 and DNAH12—dynein heavy chain 5 and 12; VPS13D—vacuolar protein sorting-associated protein 13D and NSFL1C—NSFL1 cofactor 47), cellular homeostasis, immune response (HLA-A, HLA-B and HLA-G—MHC class I antigen proteins; MAPKAPK2—MAP kinase-activated protein kinase-2 and TRAF3IPI—TRAF3-interacting protein 1), protein modifications and metabolic processes (SLC27A3—long-chain fatty acid transport protein-3 and APOL1 – apolipoprotein L1, both involved in lipid metabolism). The findings included some commonly known proteins involved in arthralgia and arthritis (i.e., MAEA; NEFL; FLG as well as COL23A1—collagen alpha 1 XXIII chain; DMGDH—dimethylglycine dehydrogenase and PDE3A—cGMP-inhibited 3′,5′-cyclic phosphodiesterase A) and also in cell death (such as USP30—ubiquitin carboxy-terminal hydrolase 30) [97].

Transcriptomics analysis of infected mice showed, as predicted, many pathways associated with viral infection with an upregulation of transcription factors involved in inflammatory response such as NFκB, USF1 (Upstream *Transcription Factor* 1), FOX (Forkhead box transcription factors), class I antigen presentation and complement pathways, cytokines and chemokines in general. Similarly, cell-death pathways were upregulated with strong apoptosis evidence along with necroptosis-related genes (such as MLKL—Pseudokinase mixed lineage kinase domain-like protein, RIPK—Receptor-interacting serine/threonine-protein kinase 1 and 3) in early and late stages of infection and BIRC2 and BIRC3 (Baculoviral IAP repeat-containing proteins) necroptosis inhibitors upregulation as well. Apoptosis genes such as caspase-3 and 9, TNF-α and pyroptosis-related genes caspase-1, IL-18 and IL-1β mRNAs showed increased expression during infection [98].

AN analysis of the *Ae. aegypti* transcriptome response to CHIKV infection showed a variety of differentially expressed transcripts from different functional categories such as binding, catalytic activity, cellular process, immune response, metabolic alterations, stimulus-response, biological process regulation, structural molecular activity, transporter activity, signal transducer activity and development in a total of 1299 upregulated and 1217 downregulated genes 3 h post-infection [99]. Orally acquired CHIKV by *Ae. aegypti* mosquitoes through a mixture of infected cells in solution revealed gene expression patterns in the midgut of the fed mosquitoes. Genes related to immunity were upregulated along with other genes not related to immune response, such as genes that encoded three synaptic vesicle protein genes, which seem to be important in viral assembly and/or budding [100], the cysteine-rich venom protein gene that contains a trypsin inhibitor-like domain (TIL) and may function as serine protease inhibitors, and two C-type lectins (CTL), which have been supposed to function as ligands, recognizing specific extracellular glycans that facilitate mosquito infection and promote gut microbiome homeostasis [101]. Other genes that were also significantly upregulated were the cytochrome p450 gene, the juvenile hormone inducible protein gene, the heat shock protein gene and a microtubule protein gene. Besides, genes that could be related to midgut escape of CHIKV when it crosses the midgut basal lamina (BL) were identified and included serine collagenases, glutamate carboxypeptidase, M1 zinc metalloproteases, serine-type endopeptidases and ten trypsins, all related to BL degradation/remodeling due to infection [102,103,104]. Comparatively, RNA-seq analysis of *Ae. albopictus* viral disseminating body parts (heads and thorax, 8 days post-infection) showed the differential regulation of various biological processes, including RNA and mRNA binding, lysosomal pathways and also the down-regulation of defensin genes, possibly as a result of either mosquito immune response or the viral modulation of immunity, which had also been observed in a previous transcriptomic analysis of *Ae. albopictus* midguts by the same authors [105]. In addition, the differential expression of two odorant binding proteins (OBPs) was observed, which could be related to reducing chemosensory and transmission mosquito abilities. Interestingly, upregulation of the BTKi, an inhibitor of Bruton’s tyrosine kinase (Btk), which is involved in several innate immune response mechanisms, was identified, and these results correlate with the effects of Btk inhibition in higher-level organisms in a protective manner, ameliorating the excessive inflammatory response [106].

Finally, NMR metabolomics of the patient’s serum analysis showed alteration in glycine, serine, threonine, galactose metabolism and TCA (*tricarboxylic acid cycle)*, similar to DENV infection [107,108]. Alteration in the TCA cycle is possible because CHIKV requires high energy for the rapid multiplication and generation of lipids, proteins, and RNA components. In addition, joint damage and arthralgia were correlated with high hypoxanthine and 4-hydroxyphenylpyruvic acid levels and carbon metabolism alterations. Sorbitol, 2-ketobutyric acid and sarcosine are biomarkers that differentiate CHIKV and DENV co-infections [107,108].

### 3.2. Mayaro Virus

The Mayaro virus (MAYV) is a neglected tropical arbovirus responsible for causing a mild febrile illness that may be accompanied by severe incapacitating arthralgia. It was first isolated from the serum of a forest worker in 1954 in Mayaro County on Trinidad Island. Since then, its presence has been reported in several countries within the tropical regions of South and Central America. It is highly underestimated as a neglected arbovirus that affects mostly poor regions in tropical and subtropical areas. Little investment has been made for epidemiological studies of it, which is considered dangerous, as it could become a public health issue due to its high urbanization potential and remarkable host plasticity [22].

MAYV comprises an enzootic cycle involving Haemagogus mosquitoes and monkeys as reservoirs. Secondary vectors such as *Ae. aegypti*, *Ae. albopictus* and *Ae. scapularis* have shown potential for viral infection and spread and secondary hosts such as humans, which poses a significant risk for urban outbreaks and epidemics [109]. The virus is, by phylogenetic analysis, a member of the Semliki complex that consists of seven other viruses: CHIKV, Bebaru virus (BEBV), Getah virus (GETV), Semliki Forest virus (SFV), Ross River virus (RRV), O’nyong-nyong virus (ONNV) and the Una virus (UNAV). The group is characterized by some common antigenic sites that generate polyclonal immune sera cross-reactivity among species and common disease manifestations such as fever, arthritis and skin rash [110]. Phylogenetic studies also classify MAYV into three genotypes: D, widely spread in South America; L, limited to Brazil; and N, recently discovered in Peru [111,112].

As a response to MAYV infection in the human hepatocyte cell line (hepG2), reactive oxygen species (ROS) were produced, causing relevant oxidative stress with increased protein carbonyl and malondialdehyde (MDA) levels, namely biomarkers of lipid peroxidation and protein oxidative modification [113]. There was also a decrease in reduced versus oxidized glutathione (GSH/GSSG) ratio and higher levels of antioxidant activity by CAT (Catalase) and SOD (Superoxide dismutase) enzymatic systems, which was not efficient in restoring the normal redox status due to high oxidative stress levels. As the infection progressed, decreased antioxidant activity and glutathione contents proved that ROS accumulation, such as H_2_O_2_, was enhanced [114]. Similarly, MAYV infection was linked to oxidative stress biomarkers in mouse models, such as malondialdehyde, carbonyl protein, myeloperoxidase (MPO) and GSH/GSSG ratio, as well as SOD and CAT activity in the liver. Liver damage was also observed by increased aspartate and alanine aminotransferases (AST/ALT) and high levels of inflammatory cells [115].

Research searching for prominent epitopes for the development of antiviral vaccines is essential, as most of these arboviruses still have no vaccine available. In a recent study for vaccine candidate epitopes conserved and homologous among CHIKV and MAYV, where identification of CD8+ T cell epitopes (CTL) was accomplished from the antigenic structural polyprotein (KPGDSGRPI, TCTMGHFIL, ALSVVTWNK, KPGRRERMC and GRRERMCMK), two other epitopes were identified for CD4+ T cells (HTL) (MCMKIENDCIFEVKH and DRTLLSQQSGNVKIT) and one for B cell (BTL) (GGRFTIPTGAGKPGDSGRPI). All epitopes demonstrated high population coverage, and screening for toxicity, allergenicity, and antigenicity yielded high safety and efficacy results. Using docking analyses, the binding affinity of HTL and CTL epitopes to HLA alleles (MHC class I and II alleles) resulted in high affinity and binding potential. However, further studies are still necessary for structural modification and stability prior to developing vaccine candidates, as these epitopes are short and could easily be degraded by host enzymes [116].

Proteomic analysis of *Ae. aegypti* Aag-2 cells infected with MAYV by label-free mass spectrometry resulted in identifying 5330 peptides and mapping several protein groups within the periods prior to and post-infection (0, 12 and 48 hpi). A total of 161 proteins were identified as differentially expressed. Proteins such as ATP synthase, heat shock proteins (HSP-20 and HSP-60), enolase phosphatase E1 (ENOPH1), membrane-related proteins (such as prohibitin, an important viral entry receptor), vesicle-associated membrane proteins (VAMPs), and overall proteins related to redox, energy metabolism, oxidative stress and viral particle maturation were identified, mostly with increased abundance over time of infection. Glycolytic enzymes related to the energy demand and manipulation of the host metabolism were identified as upregulated. An increased abundance of intermediate metabolites such as glyceraldehyde-3-phosphate was observed, demonstrating high modulation of the glycolysis pathway during infection [117].

An analysis of transcriptomic and small RNA responses to MAYV infection at 2, 7 and 14 days post-infection in *Anopheles stephensi* mosquitoes infected by blood meal resulted in 487 regulated transcripts, with 78 identified miRNAs and a siRNA response towards the MAYV genome. Enrichment of serine proteases related to the activation of the Toll pathway for an innate humoral response during infection was observed from 2 to 7 days post-infection. Moreover, later during infection, results showed the depletion of autophagic and apoptotic pathways, with induction of JNK and MPK cascades as well as repression of JAK/STAT signalling pathways due to repression of MAPK signalling, indicating that such mechanisms are part of mosquito defences against viral replication and the production of new viral particles at late stages of infection. Several differentially expressed miRNAs were identified, such as miR-286b, miR-2944a, miR-2944b, miR-307, miR-309, and novel miRNAs such as mirNOV10 and mirNOV17, and pathway analyses revealed effects in reduced oocyte viability, protein binding and signalling functions, ion channels and transport, actin filament binding and gene transcription. In addition, the identification of piRNA-like small RNAs revealed an antiviral function against replication in the mosquito, along with an increase of siRNAs during infection time related to an increased innate immune response from early to later stages of infection [118].

Bengue and collaborators’ studies on the human brain cells revealed the modulation of the immune gene expression profile in MAYV-infected astrocytes, analyzed at 48 hpi (hours post-infection) that PRRs (pattern recognition receptors) such as TLRs are present in brain cells, and the expression of TLR3, and not TLR7 (as for CHIKV infection), was found to be upregulated by MAYV, they also observed a strong induction of the PRRs IFIH1 and DDX58 transcripts by MAYV infection. The chemokines CXCL10, CXCL11 and CCL5, known to be expressed in the CNS during various viral infections, including those by encephalitic arboviruses, were upregulated by the MAYV virus, pointing to the induction of a strong inflammatory response [119].

Metabolic alterations in MAYV infected Vero cells’ exometabolome showed a variety of altered components, such as amino acids, organic acids (gamma-keto acid, beta hydroxy acid, carboxylic acid, dicarboxylic acid and phenylpropanoic acid), guanidine compounds (polyamine), monoamine (phenol), carbohydrates (monosaccharides) and lipids, all involved in glycolysis, TCA cycle, phosphate-pentose pathway or lipid β-oxidation. Therefore, alterations were observed in different periods of infection; for instance, in a 2 hpi period, metabolites such as tryptophan, 3-phenylpropionate, valine, carnitine, 3-hydroxyisobutyrate, 2-oxoglutarate and pyruvate presented high levels, whereas, in a 6 hpi period, tyramine, galactose, glucose, creatine, phosphate creatine, galactarate, serine and methyl guanidine were elevated. These results demonstrate that the alteration of metabolites directly influences biochemical reactions in response to viral infection and are generally related to amino acid metabolism, energetic metabolism and lipid metabolism that can be the focus of the future investigation of specific pathways and how viruses affect their hosts metabolically [120].

## 4. Omic’s Study across Members of Bunyavirales Order—*Orthobunyavirus* and *Phlebovirus*

### 4.1. Rift Valley Fever Virus

Rift Valley virus (RVFV) is a mosquito-borne virus known for causing Rift Valley Fever in humans and livestock throughout Africa and the Arabian Peninsula. It was first described in 1931 and isolated from inoculated lambs with infected sheep serum near Lake Naivasha in Kenya’s Rift Valley. Since then, it has caused several economically devastating epizootics with high death ratios among sheep and cattle. Outbreaks were reported in Kenya in 1968, 1978–1979 and 1997–1998, and in Southern Africa, Zambia, South Africa and Zimbabwe. In West and Central Africa, it was only isolated in 1974 and caused severe outbreaks in Mauritania in 1987 and 1998. Other countries such as Senegal, Mali, Guinea, Egypt, Tanzania, Somalia and Madagascar have also reported RVFV outbreaks throughout the years [121,122]. Transmission mainly occurs via infected mosquitoes of the genus *Aedes*, *Culex*, *Anopheles*, *Eretmapodites* and *Mansonia,* as well as *Culicoides* (biting midges), *Simuli* (black flies) and even ticks (*Rhipicephalus*). Direct contact with an infected animal’s body fluids, tissues or aerosols can also be a risk factor for the infection. Usually, veterinarians and workers who deal with livestock are at significant risk [122].

Based on molecular analysis of the Gn glycoprotein gene sequence, the classified virus isolates from 16 countries into 15 lineages (from A to O) [123]. The RVFV genome is highly conserved, with no defined serotypes and genetic differences that are mainly random single-site mutations with variable regions that are not well defined. Although differences in virulence of strains exist, genetic diversity is considerably low, and multiple strains can co-circulate during endemic periods; co-circulation, as well as co-infection with more than one viral strain, can lead to viral reassortment, a common mechanism in segmented RNA viruses, where exchange of genomic segments occurs in the host cell, resulting in either viable or non-viable reassortants [58]. Technologies for genetic surveillance are essential to tracing outbreaks’ origins and their effects. As of very recently, Juma and associates developed a web-based computational tool in order to improve RVFV genomic surveillance. The efficiency was validated by large datasets containing whole genome sequences of the L, M and S segments and the Gn glycoprotein gene sequence with optimal classification results of all 15 lineages at a phylogenetic level, proving to be an effective tool to aid in RVFV surveillance [124].

RVFV outbreaks outside the endemic regions can cause severe public health and agroeconomic issues. In humans, the disease is mainly characterized by fever. However, it can progress to a more severe state, including encephalitis, blindness, fulminant hepatitis, retinitis, thrombosis or hemorrhagic syndrome, and the mortality rates are near 20% [121]. RVFV also has a high probability of vertical transmission, as observed in Saudi Arabia in 2000, where pregnant women experienced a series of symptoms such as fever, headaches, dizziness, and myalgia and presented IgG specific to RVFV [125]. In animals, symptoms can also vary from fever, viremia, diarrhea, hemorrhage, lethargy, and death. However, there is a high chance of post-infection sequelae such as limb paralysis and severe hepatitis. In newborn lambs, the virus causes fatal disease and causes symptoms such as high fever (40–41 °C), loss of appetite, lethargy and prostration just a few hours before they die [74].

In the 1960s, a formalin-inactivated vaccine was elaborated to be tested in humans generating neutralizing antibody response [126], and ever since, several formalin-inactivated vaccines have been developed, but meagre quantities still exist and are administered and studied. In a study with seven individuals (37 to 61 years old, four males and three females) that received one to six doses of the RVFV vaccine (ranging from 7–24 years ago), T-cell responses, which have been shown to modulate and eliminate the RVFV disease, could still be detected through the ELISA and neutralization assays. This is due to the many doses of the vaccine received, although with varying responses, probably due to the many years of past vaccination. The study also presented a panel of peptides from N, Gn and Gc antigens that could be applied to further vaccination studies to investigate vaccine effects, T-cell responses and patient long-lived immune protection [127]. Other types of vaccines, such as live-attenuated and new generation vaccines, are still being investigated and developed as human and veterinary vaccines [128].

During infection, RVFV inhibits host cellular RNA synthesis through NSs protein targeting of the transcription factor TFIIH, decreasing its cellular concentration by interacting with its p44 and XPB subunits, inhibiting TFIIH subunits assembly and formation as a means of evading host response, as TFIIH is crucial for host transcription (Figure 7) [129]. Some aspects of innate and adaptive immune response to RVFV were analysed in goats. They developed viremia one-day post-infection (dpi), which persisted for at least four days. The detection of IL-6, IL-12, IL-1β, IFN-γ, and TNF-α was observed, which resulted in neutralising antibodies post viremia. The absence of IFN-α, which is one crucial antiviral cytokine, was detected [130] and, in agreement with other experiments that established that the NSs protein might also have an essential role in blocking immune response by inhibiting IFN (interferon) α/β production and action, therefore promoting evasion from host defences and presenting more virulent and pathogenic characteristics [131]. Furthermore, NSs seem to promote the downregulation of the double-stranded RNA (dsRNA)-dependent protein kinase (PKR)-mediated eukaryotic initiation factor (eLF)2a phosphorylation in order to secure efficient viral translation, as this mechanism suppresses RVFV translation when NSs is not present [132]. Another virulence factor, the RVFV NSm protein, was identified as an apoptosis inhibitor by suppressing caspase-8, a death-receptor-mediated apoptotic pathway, along with caspase-9 and caspase-3, which are downstream caspases involved in apoptotic pathways, and this makes NSm the first *Phlebovirus* protein identified with antiapoptotic function [133].

LC/MS/MS proteomic analysis of RVFV (MP-12 strain) detected various virion and host-derived protein interactions during infection. Host cytoskeleton protein association to RVFV, such as actin filaments and integrins, was identified, probably for the entry and exit of infected cells and the Golgi viral tubes formed during viral production. The Ras protein superfamily members were also involved, suggesting that they may participate in viral entry. Chaperones have also been observed from heat shock HSP70 and HSP90 protein families, such as HSPA5 (BiP) and HSPA8, subunits of the T-complex protein chaperone (TCP)—CCT2 and CCT6A, and others, working either as host factors used to promote viral replication (BiP) or with antiviral roles to prevent it (HSPA8), which makes these proteins potential targets against RVFV infection [134].

In a reverse phase protein array (RPPA) of attenuated and virulent RVFV strains infecting human cells, changes in phospho-signalling cascades were identified, especially in Smad transcription factors, which are primarily phosphorylated by and transducers of the transforming growth factor-beta (TGF-β) superfamily receptors. Increased phosphorylation of the Smad proteins was related to RVFV replication, and further analysis of transcripts altered by RVFV infection identified 913 genes that contained a Smad-response element. It was related to axonal guidance cell-signalling pathways for cellular adhesion/migration, hepatic fibrosis, cytoskeletal reorganization and calcium influx. A Smad complex on the interleukin-1 receptor type 2 (ILIR2) promoter was identified, which may be responsible for the disruption of IL-1 receptor activation and the generation of an inflammatory response during infection [135]. In human small airway epithelial cells, RVFV infection increased phosphorylation and activation of MAPK—Mitogen-activated protein kinases (with activation of p38 as a protective cellular response and control of viral replication by ERK—extracellular signal-regulated kinase) and transcription factors such as STAT1 (Signal transducer and activator of transcription 1), AFT2 (Activating transcription factor 2), MSK1 (Mitogen- and stress-activated protein kinase 1), CREB (cAMP response element binding protein) and NF-kB (Figure 8). The activation of p53 was correlated to increased levels of apoptotic pathways (caspases -3, -6 and -9) and the downregulation of antiapoptotic pathway regulator AKT (also known as PKB—protein kinase B) and apoptosis-inhibitor XIAP (X-linked inhibitor of *apoptosis* protein) [136].

Transcriptome analysis of human HEK293 cells infected with a vaccine attenuated strain of RVFV resulted in 2090 upregulated genes and 216 downregulated genes. The most upregulated genes were ISGs (interferon-stimulated genes), particularly IFIT2 (Interferon-induced protein with tetratricopeptide repeats 2) and OASL (2’-5’-Oligoadenylate Synthetase Like), both at 18 and 48 hpi, along with other 2′-5’ oligoadenylate synthase (OAS) family members and inflammatory cytokines and chemokines such as CXCL10 (C-X-C motif chemokine ligand 10), CCL5 (C-C motif chemokine 5), IL16 (Interleukin 16), ANPT2 (Angiopoietin 2) and CXCL11 (C-X-C motif chemokine ligand 11). Other upregulated genes were related to cell adhesion (ITGAM—Integrin alpha M), microtubule activity (DNAH12—Dynein axonemal heavy chain 12), Wnt signalling (FAP), membrane transport (NPC1L1—Niemann-Pick C1-Like 1), metabolism (CPA2—Cation-proton antiporter 2) and apoptosis (XAF- X-linked inhibitor of *apoptosis* (XIAP)-associated factor 1). Among the downregulated genes, MIR210HG, a microRNA involved in oxidative stress, was very relevant, possibly due to oxidative stress caused by RVFV infection, as NSs are also known to affect mitochondria function and cause redox imbalance generating reactive oxygen species, besides transcription inhibition and other virulent functions [137]. Other significantly downregulated genes were ionotropic glutamate receptors (GRIA3, GRID2), kinesin family member 12 (KIF12) and several noncoding RNAs [138].

In general, pathway analysis showed many genes involved in type I IFN response (NF-κB—nuclear factor kappa-light-chain-enhancer of activated B cells signalling, TNF—tumour necrosis factor signalling, toll-like-receptor signalling, RIG-I—retinoic acid-inducible gene-I-like receptor signalling and cytosolic DNA sensing) post RVFV infection. The key features include activating cytokine-mediated inflammatory response, cytokine-cytokine receptor interaction, chemokine signalling, and NOD-like receptor signalling. In addition, pathways involved in linoleic acid and arachidonic acid metabolism were altered, indicating their influence on immune and inflammatory responses during infection. Other pathways altered post RVFV infections were PI3K/AKT/mTOR signalling and extracellular matrix (ECM) receptor interaction, probably involved in viral entry and cell-to-cell spread of the virus [138], G2/M DNA damage checkpoint, ATM signalling, mitochondrial dysfunction and ILK signalling (also involved in cell adhesion, cytoskeletal reorganisation and cell mobility) [139,140,141].

The role of exosomes in modulating the immune response to bacterial and viral infections remains to be fully elucidated, although its importance has been recognized. How these exosomes affect viral replication in infected cells and regulate antiviral responses is still not fully clear. Alem and collaborators demonstrated that exosomes released from RVFV infected cells (Exi-RVFV) containing sequences of the viral genome but lacking the L polymerase (RdRp) or the NSs protein play a protective role against infection by inducing the RIG-I dependent activation of IFN expression leading to autophagy of infected cells in order to resist subsequent viral dissemination [142].

RNA-seq transcriptomic analysis of *Ae. aegypti* cell lines infected by RVFV and DENV (Dengue virus) revealed the upregulation of 39 genes in early (27 genes), late (22) or both early/late (10) responses. Upregulated genes from both infections and early and late responses included proteins such as PGRP (Peptidoglycan recognition protein), GNBP (Gram-negative binding protein), CTLMA 13 and 14 (C-type lectins-mannose binding) and transferrin that are involved in antiviral immune response, and family B and D clip-domain serine proteases (CLIP-B15, CLIP-B34, CLIPB-35, CLIP-B46 and CLIP-D1), which might have roles in defence responses such as hemolymph coagulation, antimicrobial peptide synthesis as well as the melanization of pathogen surfaces [143]. Comparatively, exploring *Culex pipiens* mosquitoes-RVFV molecular interactions through RNA-seq de novo transcriptomic analysis, Nuñez and collaborators identified a total of 451 differentially expressed genes (DEGs), mainly at an early stage of infection and corresponding to metabolic and cellular processes. Forty-eight were characterized by focusing on the immune response-related genes, mostly related to Toll, IMD (Immune deficiency) and RNAi (RNA interference) pathways, and a great majority were involved in apoptosis or the UPP pathway (Ubiquitin-proteasome pathway), even at an early stage of RVFV infection [144].

Metabolomic studies to evaluate the effects of RVFV infection are still not available in the literature. However, a metabolic profile of anti-RVFV medicinal plants was characterized using ^1^H NMR and chemometric analytical techniques along with UHPLC-qTOF-MS (Ultra-High-Performance Liquid Chromatography-Quadrupole Time-of-Flight Mass Spectrometry) in order to confirm the compounds that are common in plants with antiviral activity. Compounds with significant discriminating results and potential antiviral activity were selected and included chlorogenic acid, vanillic acid, ferulic acid and trigonelline, all of which have already been reported as potent antivirals. Other compounds such as flavonoids (rutin, kaempferol, flavonol glucoside and kaempferol 3-O-rutinoside) were also identified, along with two hydroxylated fatty acids (13S-Hydroxy-9Z,11E,15Z-octadecatrienoic acid and 13-Hydroxy-9Z,11E-octadecadienoic acid), which were identified for the first time in the antiviral plants’ samples [145].

### 4.2. Oropouche Virus

Oropouche virus (OROV) is the agent of oropouche fever transmitted mainly by *Culicoides paraensis* midges that circulate in South and Central America and has caused over 30 epidemics with more than half a million cases over the past 60 years in Brazil—being one of the most critical arboviruses in the Amazon region—Peru, Tobago, Trinidad and Panama. It is characterized by acute febrile illness similar to dengue fever with symptoms such as fever, headaches, skin rash, and muscle and joint pain. In severe cases, it may develop into meningitis or encephalitis [146]. In 1955, OROV was first isolated from a forest worker in Vega de Oropouche in Trinidad [147]. In Brazil, its first isolation occurred in 1960 from a blood sample of a sloth (*Bradypus trydactylus*) seized during the Belem-Brasilia Highway construction. Since then, it has circulated in the Amazon region (South and Central America) and Latin America. More recently, due to climate change, the expansion of the arthropod vectors, globalization and its potential public health significance, it has attracted research attention along with other emerging viruses [61].

OROV is a member of the *Bunyaviridae* family belonging to the *Orthobunyavirus* genus and is a member of the Simbu serogroup that comprises 22 other virus species according to serological evaluation [63]. According to genome sequence analysis, its diversity comprises four genotypes (I-IV) that are spread throughout the regions where OROV is present [148]. Genotype I was reported only in Trinidad and Tobago, while genotype II was restricted to Peru and genotype III was found only in Panama. In Brazil, both genotypes I and II were found, the former restricted to the eastern Amazon and the latest to the western Amazon region. Genotype III was found in Acre and the Minas Gerais States, and genotype IV was exclusively isolated from an outbreak in Manaus in the 1980s and is restricted to Brazil [63,149]. Genetic reassortment is also common among bunyaviruses (as in segmented RNA viruses), making genetic analyses of all three segments crucial for identifying such reassortants, which has yielded a variety of results classifying OROV reassortants based on high similarity of L, S and M segments and its proteins with at least four different viruses (Jatobal virus—JATV, Iqtos virus—IQTV, Madre de Dios virus—MDDV and Perdoes virus—PERDV) [150].

OROV infection in HeLa cells is mediated via clathrin-coated pits, and in an acidification-dependent endocytic pathway, it reaches late endosomes in about an hour. It also causes cell apoptosis via an intracellular pathway that involves mitochondria and is dependent on viral replication and protein expression [61,151]. In mice, OROV infection triggers the activation of mitochondrial antiviral-signalling protein MAVS, interferon regulatory factors such as IRF-3, IRF-7, IRF-1 and IRF-5, which regulate expression and production of type I IFN and ISGs (Interferon-stimulated genes) for host defence and viral control in non-myeloid cells [152]. High levels of liver transaminases and low glucose levels were found, which have also been reported in human serum analysis [153]. Severe liver damage was observed after OROV infection in immunocompromised mice, mainly due to high levels of cytokines and chemokines such as IL-6, IL-12, p40, G-CSF, KC and others, and immune targeting of infected cells by natural killer and/or cytotoxic T cells, as observed in Flavivirus infection [154]. In addition, IRF-5 has been suggested to play a significant role in modulating host response in peripheral organs controlling OROV dissemination in the central nervous system (CNS), inhibiting the neuroinvasive disease manifestation and also viral replication in the liver, blood, and spleen in mice in early stages of the disease [155].

During OROV infection, IFN-induced responses and TNF signalling components such as TNF-Receptor Associated Factor 3 (TRAF3) and Stimulator of Interferon Genes (STING) are essential for innate immune control. As the infection advances, the virus down-modulates these mediators, trying to escape antiviral pathways, increasing short noncoding miRNAs and type-I IFN [156]. High levels of IFN-α were present in early and late seroconverter patients, as it was considered a universal biomarker of human OROV infection. Early seroconverters presented high levels of CXCL8 (C-X-C Motif Chemokine Ligand 8) and IL-5 through the disease, consistent with the role of IL-5 in B cell activation for antibody production of plasma cells. Late seroconverters produced high levels of CXCL10 and IL-17 along with CCL2 (C-C motif ligand 2), which could be potential complementary biomarkers of OROV fever that precede seroconversion in patients [157,158]. Human peripheral blood mononuclear cells (PBMCs) were susceptible to OROV, where expression of IFNs I, II and III was increased at 24 and 48 hpi, associated with an increase in Mx1 and IFIT1 and decreased supernatant viral titers. Similarly, an increase in RIG-I and MDA5 expression was observed, which are two essential proteins for viral recognition in the cytoplasm (Figure 6), all corroborating for suppression of viral replication in these cells along with an overall increase of IFNs and ISGs, as immunodeficient environments created by blocking IFN response resulted in efficient viral replication [159].

Using sliced cultures from an adult human brain, Almeida and colleagues investigated OROV infection of mature human neural cells, which resulted in the release of TNF-α (pro-inflammatory cytokine tumour necrosis factor-alpha) and the reduction of cell viability and tissue damage in a 48-h period post-infection, which could generate further neuronal dysfunctions and neurological symptoms in human infections. Although OROV infects both neurons and microglia, the latter were the preferred central nervous system (CNS) cells infected by the virus, again demonstrating that OROV infection may lead to neuronal complications, as microglia cells are the primary defence cells in the CNS that might even serve as viral reservoirs in the brain [160]. OROV infection has also been studied in human blood mononuclear cells, and results demonstrated that T CD4+ lymphocytes (Jurkat cells) were more susceptible to infection than other lymphoid lineages. Moreover, CD3+, CD14+ and other cell populations also showed productive infection, and the presence of a viral genome throughout a few subculturing events is indicative of persistent infection [159].

Hepatocarcinoma cell line HuH-7 infected with OROV submitted to RT-qPCR-based screening detected 13 differentially expressed miRNAs, known for regulating gene expression post-transcriptionally and playing important roles in viral infections [161]. These results demonstrated that miR-217 and miR-576-3p were upregulated during infection, and gene targets involved in apoptosis, type I interferon-mediated response, and antiviral restriction factors were associated with those miRNAs. Also, miR-217 and miR-576-3p are described as inhibitors of IFN-β antiviral response, leading to a decrease in antiviral response as the infection continues, and the NSs protein regulates innate host immunity response, modulating the IFN pathway [156,162].

An investigation of genome-wide transcriptomic alterations of human primary astrocytes infected by the Oropouche, Zika, Chikungunya and Mayaro viruses presented different pathways affected by each virus, but with common signatures related to the down-regulation of innate immune pathways, antiviral response and interferon-associated inflammatory cytokines. Examples of common down-regulated genes/protein expression were DDHX58 (RIG-I), IFN- β and transcription factors related to interferon pathways such as IRF-7. OROV infection alone revealed the up-regulation of pathways related to cell division and cell cycle, DNA maintenance and replication, along with extracellular matrix remodeling (SPARC/Osteonectin/SPOCK3; ITGAM—Integrin Subunit Alpha M; and COL17A1—Collagen Type XX Alpha 1 Chain; COL20A1—Collagen Type XX Alpha 1 Chain). Interference with cell cycle, DNA condensation and cellular transcription has been observed with orthobunyaviruses. Cell cycle arrest might account for the increased availability of capped mRNAs to act as donors for translation, the cap-snatching mechanism [163,164]. The down-regulation of interferon response and genes with antiviral activities was also observed (RSAD2—Viperin; OASL and OAS1—2′5′-Oligoadenylate Synthethase, MX1/2—MX Dynamin-like GTPase 1; IFIT2—Interferon-induced Protein with Tetratricopeptide Repeats 2), and evidence of the down-modulation of ion channels, transmembrane solute carriers and synapse regulation pathways was also identified, revealing manifestations of OROV neuropathology [165].

As no vaccination is available, studies on immunogenic and antigenic epitopes are fundamental for developing antiviral therapies. Through immunoinformatic techniques, analyses of OROV proteome for the search of T-cell and B-cell epitopes were performed along with finding inhibitor-binding sites by docking simulation of the putative OROV polyprotein, which was identified as the most antigenic protein with a total of 128 epitopes with the highest antigenic scores. By narrowing criteria (antigenicity, immunogenicity and conservancy), a total of 18 highly conserved T-cell epitopes were identified as potential vaccine candidates, with significant population coverage and with one particular epitope (LAIDTGCLY) defined as a putative vaccine candidate due to its binding affinity to both MHC-I and MHC-II alleles identified through docking experiments. Moreover, five highly conserved B-cell epitopes were also identified, and one in particular (HHYKPTKNLPHVVPRYH) presented promising results. The 3D OROV polyprotein structure presented ten possible ligand-binding pockets, and its amino acid residues were identified, supporting its potential as an antiviral target for future research against OROV infection [166].

## 5. Remarks and Conclusions

As arboviruses are the causative agents of recurrent epidemics worldwide, and the infection of humans may result in long-term complications and chronic conditions, understanding the viral structure, mechanisms and interactions with hosts is a key step for the development of antiviral approaches. Among the nearly 540 arboviral species comprised within several families and genera are the mosquito-borne Chikungunya, Mayaro, Oropouche and Rift Valley viruses which belong to the *Alphavirus*, *Orthobunyavirus* and *Phlebovirus* genera, respectively, and which were the main focus of this review, which aims to provide information on arboviral mosquito infection-related aspects obtained from "omics" studies and approaches. Herein, we aimed to display how some of the tools and technology sensitivity of omic approaches provide a better understanding of biological systems and validate the potential effects these viruses have on experimental animal models, cell culture, and, more importantly, humans. The study integration of genes to transcripts, proteins and metabolites can provide a holistic and unique perspective on biomarkers and biochemical fingerprints present in a specific host due to different conditions set by pathogens or diseases. Improving knowledge of how these viruses affect organisms, especially humans, is critical for developing future therapies or medications to prevent or treat such relevant public health issues. Although the combination of several omics approaches and the integration of these results for biological interpretation is still a challenge, thanks to technology advances and improvement of online databases, future studies regarding these viruses or other pathogens under the multi-omics approach might be able to offer new perspectives on systems biology to elucidate several aspects of such recurrent human pathogeneses.

## Figures and Tables

**Figure 1 viruses-14-02194-f001:**
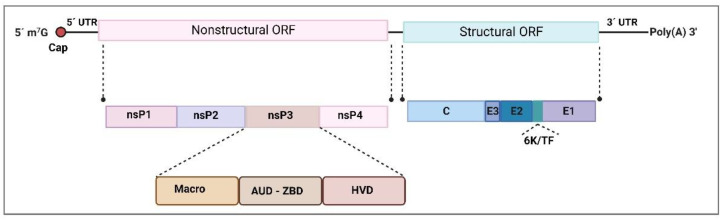
*Togaviridae* family. Genome structure of *Alphavirus* and nsP3 non-structural protein domains (the macro domain, the alphavirus unique domain (AUD) and the hypervariable region (HVD). The genome organization can be summarized as 5′m^7^G Nonstructural ORF-(junction)-Structural ORF-An3′, in which two open reading frames (ORFs) are present. The non-structural ORF (NS-ORF) encodes the four non-structural proteins (nsP1-nsP4) which are required for replication. The structural ORF, produced by a subgenomic mRNA, encodes the structural proteins in the viral particle (C, E3, E2, 6K/TF, E1). The genomic RNA with noncoding regions is represented as solid black lines and ORFs as open boxes. Based on Götte et al. (2018) [24] and created with BioRender.com (Accessed on 14 September 2022).

**Figure 2 viruses-14-02194-f002:**
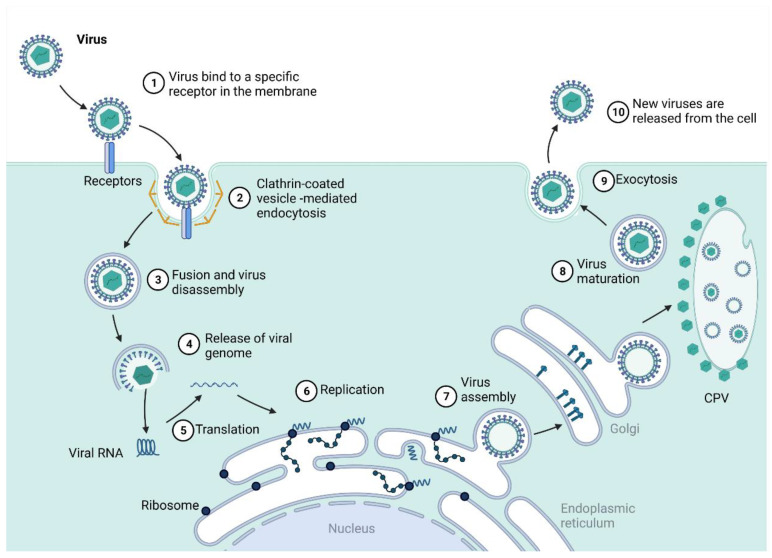
*Alphavirus* replication. After binding to specific receptors in cells (1), membrane invagination occurs, forming clathrin-coated vesicles with the viral particle inside (2) that are later taken by endosomes, whose low pH environment promotes envelope proteins E1/E2 fusion to the endosomal membrane, and the nucleocapsid is released into the cytoplasm with further viral RNA liberation (3, 4). Translation of the non-structural proteins is initiated, and replication occurs associated with ER membranes forming the cytopathic vacuoles (CPVs) (5, 6). After the replication process and synthesis of structural proteins, viral assembly and maturation (7, 8) take place, and budding from the cell membrane occurs through interactions of the nucleocapsid proteins with membrane glycoproteins, and virion are released to infect new cells (9, 10). Based on: Garoff, Sjôberg and Cheng (2004) [32] and created with BioRender.com (Accessed on 14 September 2022).

**Figure 3 viruses-14-02194-f003:**
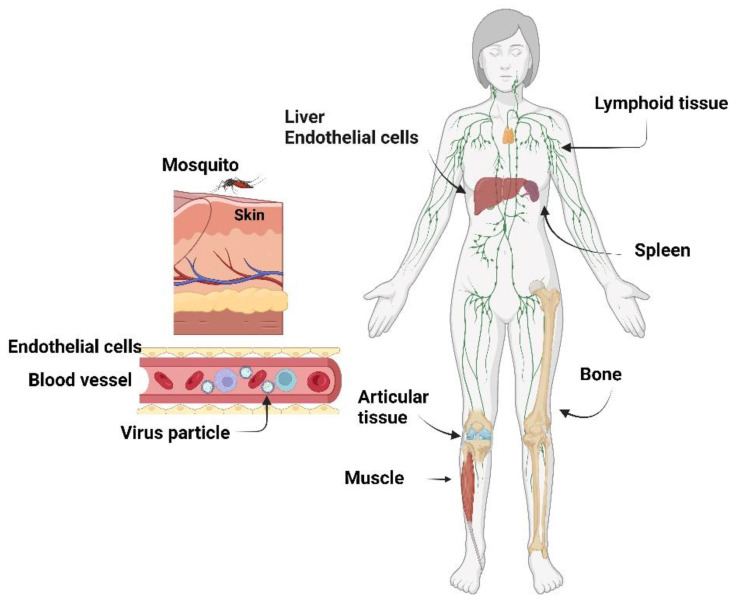
*Alphavirus* dissemination. Probable alphavirus dissemination in humans after inoculation of the virus through the mosquito vector bite. It propagates into the target tissues (muscle, articular tissue, liver, spleen, etc.) via the blood vessels. Created with BioRender.com (Accessed on 10 September 2022).

**Figure 4 viruses-14-02194-f004:**
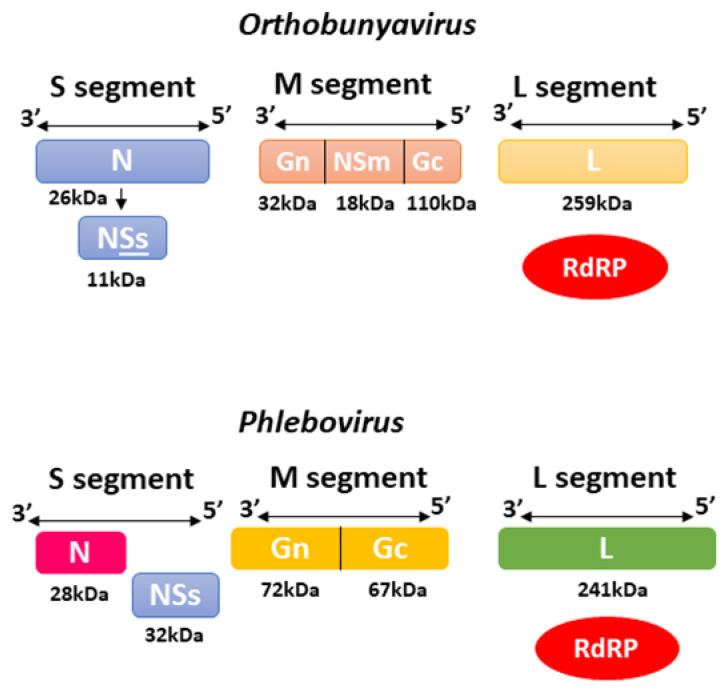
Genome segments of *Orthobunyavirus* and *Phlebovirus*. The three-segmented negative-sense single-strand RNA are nominated according to their size as L (large), M (medium) and S (small). The L segment encodes the multifunctional RNA-dependent RNA polymerase (RdRP) named L protein. The medium segment encodes the Gn and Gc glycoproteins, and the small segment encodes the nucleocapsid protein (N). Additional non-structural proteins such as NSm and NSs can also be encoded by segments M and S, respectively, which can be cleaved by cellular proteases during polyprotein maturation. Based on: Ferron et al. (2017) [65].

**Figure 5 viruses-14-02194-f005:**
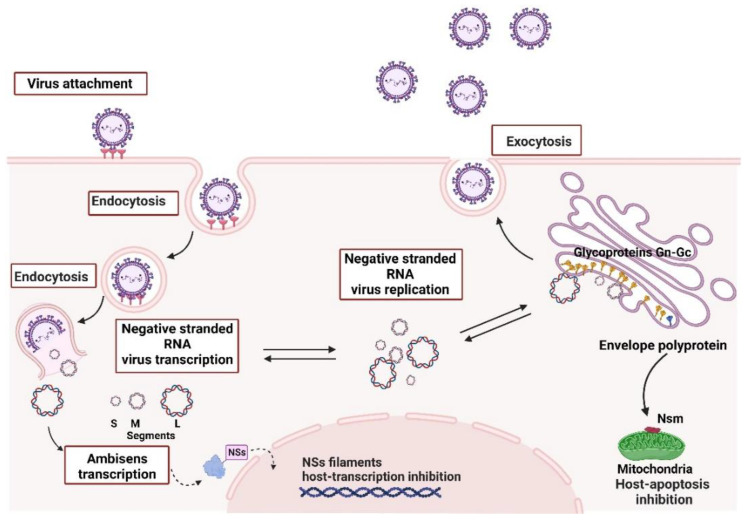
Replication process of *Orthobunyavirus* and *Phlebovirus***.** The virus enters the cells through clathrin-mediated endocytosis, and endosome membrane fusion occurs at acidic pH facilitated by viral glycoproteins, uncoating and liberating the viral RNA into the cytoplasm. The viral polymerase L and the N protein catalyze transcription and replication of the viral genome in the cytoplasm. Translation and transcription are coupled processes due to the action of translocating ribosomes, which are able to prevent nascent mRNA interactions and premature termination, allowing transcription to occur up to the termination signals at the 5′ end. In replication, antigenomic RNAs (cRNAs) are synthesised and used as templates for progeny genomic RNAs (vRNAs). The non-structural NSs and NSm proteins have been proposed to shut down host transcription and apoptosis for efficient viral replication. Interaction of the viral glycoproteins Gn and Gc in the Golgi apparatus allows packaging of the RNP complex into the viral particles. Further modifications and maturation steps occur until the new virions are released from the cells. Based on: Guu, Zheng and Tao (2012) [72]. Created with BioRender.com (Accessed on 11 September 2022).

**Figure 6 viruses-14-02194-f006:**
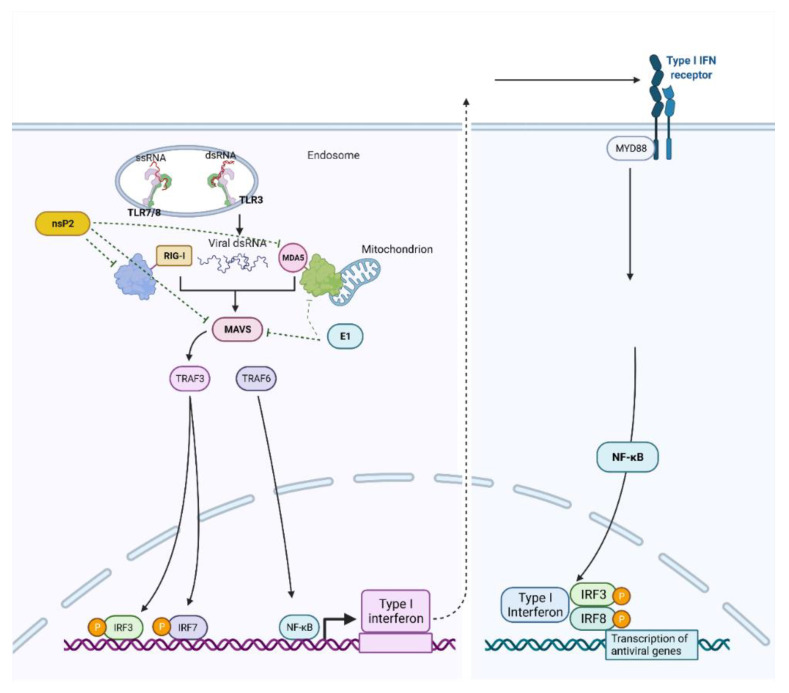
Proposed CHIKV signalling cascades. Engagement of the pathogen receptors in cells (TLR-X) and (RIG-I) induces a cascade that activates the type I interferons. Recognition of viral RNA in the endosomes of infected cells leads to the recruitment of adaptor proteins such as MyD88. This results in activating several signalling cascades and the phosphorylation of transcription factors: IRF3, IRF7 and NF-kB (Nuclear factor kappa B). When phosphorylated, these transcription factors can translocate from the cytosol into the nucleus and induce the transcription of type I interferons (IFN-α and IFN-β) and proinflammatory cytokines (IL-1b, IL-6, TNF-α). CHIK nsP2, E2 and E1 proteins suppress IFN-β through the MDA5-RIG-I pathway. Created with BioRender.com (Accessed on 11 September 2022).

**Figure 7 viruses-14-02194-f007:**
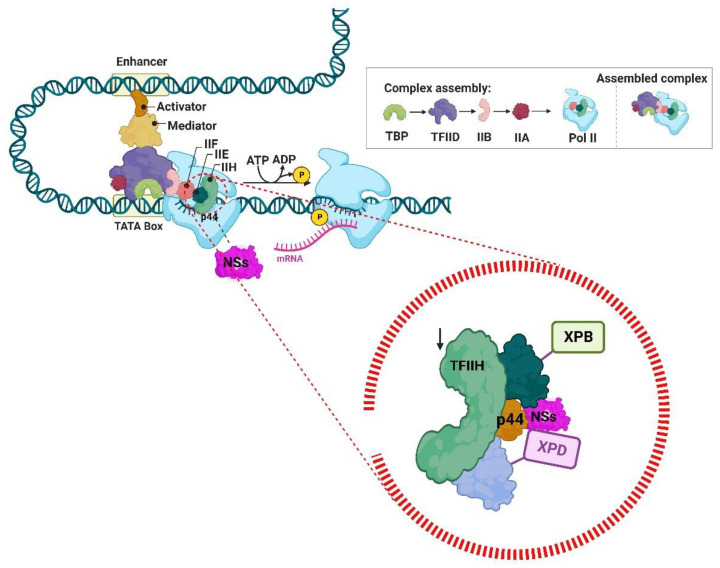
Schematics of RVFV NSs-mediated TFIIH suppression. The top portion illustrates the transcription factor of the RNA. At the same time, NSs protein (pink) binds to p44 and XPB, inhibiting TFIIH subunits assembly and the formation of the complex to evade host response and shut off host transcription. Created with BioRender.com (Accessed on 14 September 2022).

**Figure 8 viruses-14-02194-f008:**
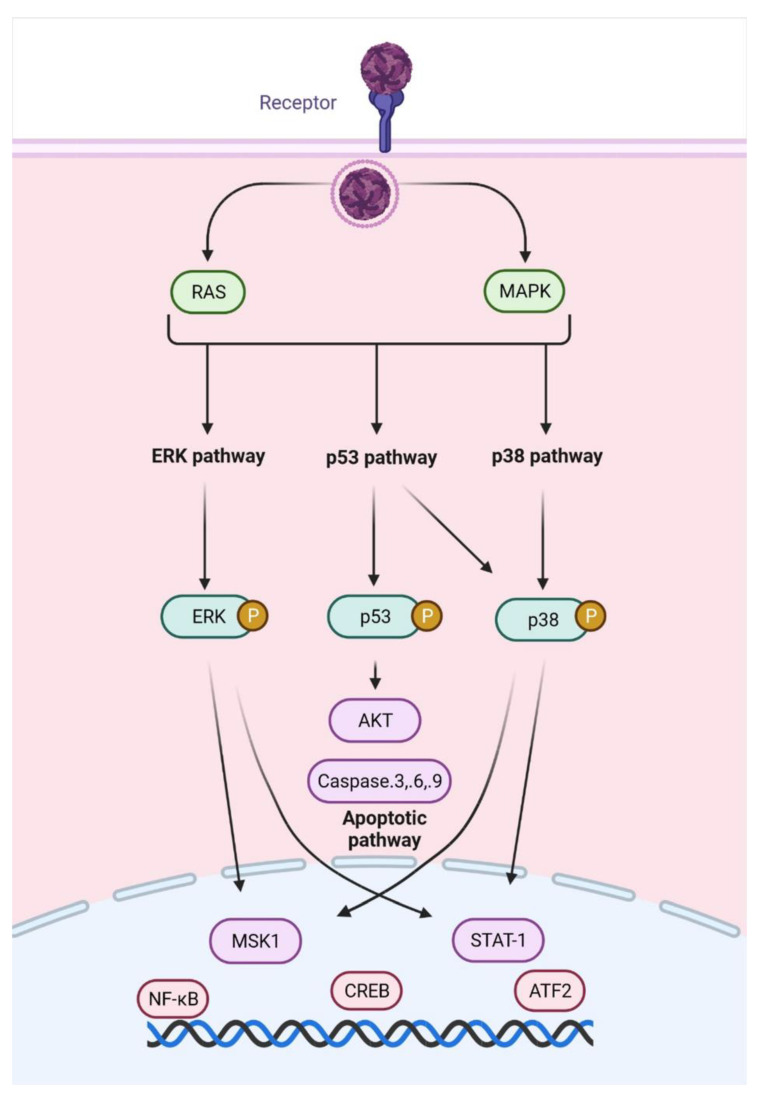
RVFV infection leads to the induction of ERK, p38, MAPK, and p53 phosphorylation. The activation of p53 escorts transcriptional upregulation of the apoptotic genes correlated with the increased levels of cleaved effector caspase -3, -6 and -9 and apoptotic pathways. Created with BioRender.com (Accessed on 13 September 2022).

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
