# Peer review of "A Review of Omics Studies on Arboviruses: Alphavirus, Orthobunyavirus and Phlebovirus"

_viruses, 2022, doi:10.3390/v14102194_

Round 1

Reviewer 1 Report

The manuscript represents a comprehensive overview of the current knowledge regarding structure and replication of Arboviruses, focussing on 3 different families. The same authors have very recently published a quite similar review in Virus Research, focussing on the Flavivirus genus of Arboviruses (published in 2022). The review reads ok, some phrasing issues need to be corrected. My literature research shows that the information was not compiled in a similar way within the past years, thus it seems to be reasonable and helpful for the community to publish such a review. However, the summarized literature does not contain too much papers of the past 3 years, which points into the direction that maybe more new literature could have been included. But I could not check it so far.  Moreover, its without doubt in the scope of the Journal Viruses.

Thus, I think its worth to be published, if the citeations are adequate in the contet completely fits, which is hopefully commented on by the other two reviewers.

Reviewer 2 Report

In the manuscript entitled “Brief review of Omics tools in the study of Alphavirus, Orthobunyavirus and Phlebovirus” by Rafaela dos S. Peinado et al, the authors review lifecycles, known interactions between virus and host proteins, and histories/health data for Alphaviruses, Orthobunyaviruses, and Phleboviruses; they then summarize omics related data generated for representative viruses in these families. While the review title reflects a timely subject highly worthy of review, this manuscript could be substantially improved.

Major issues:

1)    The title and abstract suggest this review will be focused on omics, yet omics studies aren’t even brought up until page 10. Figures do not incorporate any of the recent omics data, and omics tools aren’t discussed, rather a summary of result from omics-based studies.

2)    Authors spend a great deal of space on virus histories and health impacts, which don’t fit with the objectives of the paper. Overall, the “brief” review is only brief in relation to the subject of the review.

3)    No conclusion or future directions are presented because of the omics studies, and the results aren’t tied back to the extensive introduction before reaching the omics sections. This review needs to be re-written to emphasize omics or be re-branded to fit what is reviewed.

Minor issues:

There are extensive spelling errors that should be easily caught by a spell checker, and many randomly included double spaces.

Figure legends need to be proof-read.

Figure 1: E1 blue is too dark making E1 nearly unreadable.

Figure 3: Match the text sizes, “Lynphoid” spelled wrong, and in the figure legend what is “Scheme 4”, “symptons” should ge “symptoms”?

Many run-on sentences that are difficult to interpret meaning from.

Line 170: RIG-I not RIG-1

Line 192: “TATase” should be “ATase”

Line 195: “P123” and “P1234” aren’t defined. There are many of these types of introductions in the manuscript without context.

Figure 5: Make the words in the figure larger, many spelling errors in legend.

Figure 7: “Complex assembly” inset is tiny, make readable size.

Figure 8 increase size.

Reviewer 3 Report

The review article is well-written and gives a comprehensive overview to the state of knowledge in the field including beautiful figures. I do not have specific points for improvement. 

Round 2

Reviewer 2 Report

In the manuscript entitled “A review of Omics studies on arboviruses: Alphavirus, Orthobunyavirus and Phlebovirus” by Rafaela dos S. Peinado et al, the authors review lifecycles, known interactions between virus and host proteins, and histories/health data for Alphaviruses, Orthobunyaviruses, and Phleboviruses and summarize omics related data generated for representative viruses in these families. The review reflects a timely subject worthy of review, and the renewed title now far better matches the content of the manuscript. The reviewer is satisfied with the improvements.

Author Response

Thank you for the positive comments!